# The Study of the Applicability of Electron Irradiation for FeNi Microtubes Modification

**DOI:** 10.3390/nano10010047

**Published:** 2019-12-24

**Authors:** Daryn B. Borgekov, Maxim V. Zdorovets, Dmitriy I. Shlimas, Artem L. Kozlovskiy, Kayrat K. Kadyrzhanov

**Affiliations:** 1Engineering Profile Laboratory, L.N. Gumilyov Eurasian National University, Astana 010008, Kazakhstan; borgekov@mail.ru (D.B.B.); mzdorovets@inp.kz (M.V.Z.); shlimas@mail.ru (D.I.S.); kayrat.kadyrzhanov@mail.ru (K.K.K.); 2Laboratory of Solid State Physics, The Institute of Nuclear Physics, Almaty 050032, Kazakhstan; 3Department of Intelligent Information Technologies, Ural Federal University, 620075 Yekaterinburg, Russia; 4Department of Arts and Humanities, Information and Communication Technologies, Services and Engineering, Manufacturing and Construction Industries, Kazakh-Russian International University, Aktobe 030006, Kazakhstan

**Keywords:** nanostructures, modification, electronic annealing, vacancies, phase composition

## Abstract

The paper presents the results of a study of irradiation of high-energy electrons by an array of FeNi nanostructures with doses from 50 to 500 kGy. Polycrystalline nanotubes based on FeNi, the phase composition of which is a mixture of two face-centered phases, FeNi_3_ and FeNi, were chosen as initial samples. During the study, the dependences of the phase transformations, as well as changes in the structural parameters as a result of electronic annealing of defects, were established. Using the method of X-ray diffraction, three stages of phase transformations were established: FeNi_3_ ≅ FeNi→FeNi_3_ ≪ FeNi→FeNi. After increasing the radiation dose above 400 kGy, no further phase changes were followed, indicating the saturation of defect annealing and completion of the lattice formation process. It was found that an increase in the degree of crystallinity and density of the microstructures as a result of irradiation indicates electronic annealing of defects and a change in the phase composition. It was established that the initial microtubes, in which two phases are present, leads to the appearance of differently oriented crystallites of different sizes in the structure, which contributes to a large number of grain boundaries and also a decrease in density, and are subject to the greatest degradation of structural properties. For modified samples, the degradation rate decreases by 5 times. In the course of the study, the prospects of the use of electron irradiation with doses above 250 kGy for directed modification of FeNi microtubes and changes in structural features were established.

## 1. Introduction

Important conditions for the applicability of nanostructured materials and devices based on them is an increase in operational characteristics, chemical and thermal stability, and resistance to external influences, which has a large role in the life of nanostructured materials [1,2,3]. In this regard, one of the urgent tasks in nanostructured material science is to study the evolution of physicochemical properties, corrosion resistance, and the degradation of the structure of nanomaterials as a result of external influences, such as thermal heating, exposure to aggressive media, irradiation with ionizing radiation, etc. [4,5,6]. The presence of defects, residual stresses, and the nonequilibrium phase states in nanostructures leads to the formation of excess free energy while the external effect on the nanostructure plays a dual role. On the one hand, it can initiate the processes of recrystallization, relaxation, and segregation, leading to changes in nanostructures, and on the other hand, accelerate the degradation of nanostructures [7,8]. From this point of view, new knowledge of the stability of nanostructures to external influences is of great importance for both fundamental and applied research, which underly the development of methods for determining the reliability of nanostructures and assessing their applicability [9,10,11,12]. The stability of the physical and structural properties of nanomaterials is the basis for determining the reliability of devices based on them. One of the factors leading to changes in physical properties, such as electrical and structural ones, is controlled modification by irradiation of nanostructures. Radiation effects arising in nanomaterials as a result of exposure to ionizing radiation have a number of features that differ from similar effects in micro- and macro-sized objects [13,14,15]. In turn, the presence of a large number of grain boundaries and junctions, which are sinks for removing radiation defects, enhances the stability of nanomaterials compared to bulk materials [16,17]. The use of various types of ionizing radiation for the modification of nanostructures is an effective tool for changing the physicochemical properties of nanostructures for studying radiation resistance processes. In this regard, it is of great interest to study the effect of ionizing radiation on the structural and conductive properties of nanotubes, as well as to assess the possibility of using ionizing radiation to increase the corrosion resistance and change the conductivity of nanostructures [17,18,19,20,21].

One of the promising classes of nanomaterials for microelectronics and magnetic devices are structures based on iron-nickel alloys. It is known that among iron-nickel compounds, the best characteristics are possessed by Invar (Fe_64_Ni_36_) and Permalloy (Fe_20_Ni_80_) alloys, which have high corrosion resistance, good conductivity, and magnetic permeability, which makes them indispensable in the manufacturing of various microelectronic devices, magnetic sensors, and anti-corrosion coatings exposed to aggressive environments and external environmental factors [22,23,24,25,26,27]. Moreover, in nanostructured materials, these properties are more pronounced, since the size effect plays the main role in their change. In this connection, it is of great interest to study the processes of defect formation, as well as the possibility of using ionizing radiation to modify the properties of iron/nickel-based nanostructures.

The paper presents the results of a study of the applicability of electron irradiation to modify the structural properties and phase composition of FeNi-based microstructures. These microstructures were obtained using the template synthesis method. The interest in this class of microstructures is due to the possibility of their use in microelectronics, catalysis, and biomedicine. To modify the structural properties, an electron radiation with an energy of 5 MeV was chosen, which allows electronic annealing of defects over the entire volume of microstructures.

## 2. Materials and Methods

### 2.1. Electrochemical Synthesis of Nanostructures

The synthesis of nanostructures of a given composition was carried out using the method of electrochemical synthesis, which is based on the reduction of metal ions in a template matrix from electrolyte solutions under the influence of an electric current. Polymeric track membranes based on polyethylene terephthalate (Hostaphan trend, Mitsubishi Polyester Film Corp., Wiesbaden, Germany), obtained using ion-track technology and subsequent chemical etching of the tracks to the desired geometry, were used as template matrices for nanostructures. The essence of the ion-track technology method is to irradiate with heavy Kr^14+^ ions with an energy of 140 MeV and a fluence of 10^7^ ion/cm^2^ polymer films with a thickness of 12 µm in order to obtain latent tracks, which were subsequently etched to a diameter of 400 nm in a solution of 2.2 M sodium hydroxide at a temperature of 85 ± 1 °C. Before etching, polymer films are pretreated with UV radiation in order to embrittle latent tracks and accelerate the etching process of tracks [28,29].

For the electrochemical synthesis of nanostructures, a two-electrode cell with copper cathodes was used, the distance between which is fixed. The process of the formation of nanostructures in polymer templates was monitored using the method of chronoamperometry, which consists in controlling the increase in the current density as the pores of the template matrices are filled. The difference in the applied potentials for deposition was 1.75 V. As the electrolyte solution, we used iron and nickel salts—FeSO_4_ × 7H_2_O, NiSO_4_ × 7H_2_O in an equal molar ratio. The addition of boric (H_3_BO_3_) and ascorbic (C_6_H_8_O_6_) acids was used to achieve the required solution pH of 3, and also as buffer compounds to accelerate the crystallization of crystallites on the walls of tracks in polymer matrices [30,31]. To form nanostructures in the pores of the template matrices, a layer of gold 30 to 50 nm thick was deposited on one side by magnetron sputtering, which served as the cathode during deposition. The geometry of the deposition of the conductive layer was designed in such a way that activation centers consisting of gold are formed on the pore walls near the edge of the polymer, which subsequently serve as nucleation centers at the initial stages of the formation of nanostructure walls.

### 2.2. Study of Morphology, Structural Characteristics, and Phase Composition

The morphological features of the synthesized nanostructures were studied using the method of scanning electron microscopy using the “Hitachi TM3030” instrument (Hitachi Ltd., Chiyoda, Tokyo, Japan). The study of the elemental composition, as well as the mapping of the structures under study in order to determine the equiprobable distribution of elements in the structure, was carried out using the energy dispersive analysis method performed using the EDA Bruker Flash MAN SVE installation (Bruker, Karlsruhe, Germany). The phase composition was studied, as well as the dynamics of changes in the X-ray diffraction patterns of the studied samples before and after irradiation were evaluated using the X-ray diffraction method performed on a D8 ADVANCE ECO powder diffractometer (Bruker, Karlsruhe, Germany). Conditions for recording diffractograms: 2*θ* = 20–80°, step 0.02°, spectrum acquisition time 3 s, X-ray radiation Cu-Kα, λ = 1.54 Å. An analysis of the structural characteristics, as well as the phase composition, was carried out using TOPAS v.5.0 software based on the Rietveld method [32,33].

To estimate the parameters of the crystal lattice, the Nelson–Taylor extrapolation function was applied (Equation (1)):(1)$$a=f\left[\frac{1}{2}\left(\frac{\cos^{2}\theta }{\sin\theta }+\frac{\cos\theta }{\theta }\right)\right].$$

The value and the error in determining the parameter a are determined by linear extrapolation of this function to the zero value of the argument (*θ* = 90°).

An analysis of the angular dependence of physical broadening allows one to evaluate the influence of both factors. To assess the impact, the Williamson–Hall method was used, which is based on Equation (2):(2)$$\begin{array}{l} \beta^{2}=W_{size}^{2}+W_{strain}^{2}\\ W_{size}^{2}={\left(\frac{\lambda }{D\cdot \cos(\theta )}\right)}^{2}\\ W_{strain}^{2}={\left(4\cdot \varepsilon \cdot \tan(\theta )\right)}^{2} \end{array},$$
where *β* is the physical broadening of the diffraction maximum, λ is the X-ray wavelength (1.54 Å), *D* is the crystallite size, *θ* is the Bragg diffraction angle, *ε* is the magnitude of microstresses in the lattice.

Microstresses were estimated based on an analysis of the displacement of diffraction peaks calculated according to Equation (3): (3)$$microstrain=\frac{d_{irr}-d_{pristine}}{d_{pristine}},$$
where *d_pristine_*, *d_irr_* are the interplanar distances before and after irradiation.

The volume fraction of the phase contribution was determined by evaluating the change in the shape of the diffraction lines and their intensities using Equation (4): (4)$$V_{admixture}=\frac{RI_{phase}}{I_{admixture}+RI_{phase}}$$
where *I_phase_* is the average integral intensity of the main phase of the diffraction line, *I_admixture_* is the average integral intensity of the additional phase, and *R* is the structural coefficient equal to 1.45.

The density of the studied microtubes was calculated using Equation (5): (5)$$p=\frac{1.6602\sum AZ}{V_{0}}$$
where *V*_0_ is the volume of the hexagonal cell, *Z* is the number of atoms in the crystal cell, and *A* is the atomic weight of the atoms.

The degree of disorder of the crystal lattice of the studied samples was found according to Equation (6): (6)Pdil=(1−pp0)∗100%,
where *p*_0_ is the density of the reference sample.

The dislocation density (*δ*) contains information on the improvement of the crystal structure and is calculated according to Equation (7):(7)$$\delta =\frac{1}{D^{2}},$$
where *D* is the crystallite size.

The concentration of vacancies was calculated using Equation (8):(8)$$\beta =\frac{\left(a_{th}^{3}-a_{ex}^{3}\right)}{a_{th}^{3}}*100\%,$$
where *a_th_* and *a_ex_* are the reference and experimental values of the crystal lattice parameter.

The ultrafine magnetic field parameters of the synthesized microtubes were studied using the Mössbauer spectroscopy method (MS1104Em spectrometer, RAS LLC, Rostov on Don, Russia). Processing and analysis of the obtained Mössbauer spectra was carried out taking into account a priori information on the structural and dimensional characteristics of the studied objects. For processing, SpectrRelax software was used.

### 2.3. Directional Modification of Nanostructures

The structural properties and phase composition of the synthesized nanostructures were modified using an ELV—4 linear accelerator (Park of nuclear technology, Kurchatov, Kazakhstan) by irradiating an irradiation dose of 50 to 500 kGy with a step of 50 kGy with a 5 MeV electron beam. The studied nanostructures were irradiated in air. Dose control was carried out using film detectors. According to the calculated data, at given electron energies, their mean free path in a similar material is more than 20 µm, while the length of nanostructures is 12 µm. In this case, we can talk about modifying the properties of nanomaterials along the entire length. According to the calculated data, as a result of collisions of electrons with an energy of 5 MeV, the maximum energy transferred to the atom in an elastic collision (T_MAX_) = 250 eV, which leads to the formation of single defects, since the energy of the primary knocked-out atom is not enough to create cascades of secondary defects. This circumstance served as the basis for the use of electron radiation for radiation annealing of defects in the crystal structure of materials [34,35].

## 3. Results

Figure 1 shows an SEM image of an array of synthesized microstructures without a polymer matrix, which was removed by chemical etching of the polymer in a highly concentrated sodium hydroxide solution at a temperature of 50 °C for 30 min. These etching conditions make it possible to completely dissolve the polymer without damaging the structure of the microtubes. Before etching, one of the sides was coated with a metal substrate 0.5 to 0.6 µm thick to hold the microstructures in the form of an array. As can be seen from the data presented, the geometry of the microstructures completely repeats the geometry of the template tracks while the uniformity in the height of the synthesized structures indicates that the formation of microstructures is the same throughout the template. The strength properties of the microstructures were estimated by counting the deformed and broken microtubes in the statistical analysis of SEM images of the microstructure arrays. As a result, it was found that the proportion of deformed structures is 0.5% to 1% of the total number of microwires. It is worth noting that a detailed analysis of the obtained microtubes did not reveal visible defects in the form of cracks or hollow regions on the surface, which indicates a high degree of crystallinity and good strength properties of the obtained microstructures.

Figure 1b,c presents the energy dispersive analysis data and the results of the mapping of the samples under study. According to the data obtained, the elemental iron/nickel ratio in the structure of the microtubes is 61/39 at.%, which is typical for Invar-type iron/nickel alloy compounds. It should be noted that the obtained value is the average value, which was calculated by evaluating the elemental composition along the entire length of the microtubes at various points. To assess the uniformity of the distribution of elements in the structure, a mapping method was used, according to which it was established that in the structure of microtubes, there are regions with a high nickel content, the presence of which may be due to the formation of metastable phases containing large amounts of nickel. Moreover, the higher iron content in the structure is due to the lower potential for the reduction of iron ions from solution, as a result of which the rate of reduction of iron exceeds the rate of reduction of nickel. Different amounts of ion recovery can lead to the formation of various phases in the structure, as well as the appearance of additional distortions and deformations in the structure, which can have a negative effect on the properties of microstructures. Figure 1d shows the Mössbauer spectrum and the result of reconstructing the distribution of the hyperfine magnetic field of the test sample obtained at room temperature. In the general case, the spectrum of the studied sample is a Zeeman sextet with broadened lines characteristic of the FeNi structure and two quadrupole doublets characteristic of Fe^3+^ impurity cations in the microtube structure. The presence of impurity inclusions, as well as the broadening of the lines of the partial spectrum, indicates the disorder of the crystal and magnetic structure, which is due to distortion and deformation of the structure during the synthesis.

Figure 2a shows the dynamics of changes in the X-ray diffraction patterns of the studied samples before and after irradiation. According to the obtained X-ray diffraction data, the initial microstructures are a mixture of two phases of FeNi_3_ and FeNi from a face-centered type of crystal lattice. The presence of two phases in the structure, as mentioned above, is due to the processes of the formation of microstructures in the process of electrochemical reduction of metal ions from an electrolyte solution. Moreover, an equal concentration of salts during the preparation of the electrolyte solution leads to an equally probable concentration of Fe^2+^ and Ni^2+^ ions in the solution. However, different values of the ion reduction potentials lead to a slight dominance of the iron ion reduction rate, as a result of which the iron content in the structure is more than 60 at.%. As a result, when the crystal structure is formed, two face-centered phases of different spatial syngonies Fm-3m(225) for the FeNi phase and Pm-3m(221) for the FeNi_3_ phase are formed, with close crystal lattice parameters, a = 3.5361 Å for the FeNi phase and a = 3.5293 Å for the FeNi_3_ phase. According to the estimation of the areas of diffraction peaks and their contributions to the diffraction pattern using the Rietveld method, the ratio of the FeNi/FeNi_3_ phases was established, which amounted to 46.5/54.5%. Assessment of the shape and width of the diffraction peaks made it possible to estimate the average crystallite size for various phases, which was 12.7 ± 1.8 nm for the FeNi_3_ phase and 17.5 ± 2.1 nm for the FeNi phase. Slight differences in crystallite sizes indicate small differences in phases while a small crystallite size leads to a high density of dislocation defects, which is due to the presence of two phases, as well as a large number of grain boundaries.

It is worth noting that the appearance of the diffraction pattern of the initial sample indicates the presence of distortions and deformations in the structure, as evidenced by the asymmetric shape of the diffraction peaks, as well as the shift of the peaks toward small angles. As is known, two factors influence the change in the shape and width of the diffraction lines: The size factor, associated with the change in crystallite size, and the strain factor, associated with distortions and deformation of the interplanar spacings. One of the methods for separating these contributions is the Williamson–Hall method, which allows one to assess the degree of deformation and distortion of the crystal structure. Figure 2b shows the construction of the angular dependence of the change in the width at half height (FWHM) of the diffraction peaks for the studied samples. For the initial samples and samples irradiated with doses of 50 to 250 kGy, the FWHM value is divided into two regions characteristic of different phases of FeNi_3_ and FeNi. The slope of the curve characterizes the magnitude of the micro-distortions of interplanar spacings and deformations of the structure. It can be seen from the presented dependences that an increase in the radiation dose leads to a decrease in the angle of inclination, which indicates a decrease in the deformation contribution to the distortion of the crystal structure. In this case, for samples irradiated with doses of 300 to 500 kGy, for which the presence of one phase in the structure is characteristic, the slope of the curve is minimal, which indicates a small contribution of distortions of the crystal structure. Figure 3 shows the dependences of the change in micro distortions and crystallite sizes in the structure of microtubes according to the Williamson–Hall method. As can be seen from the data presented, the largest change with an increasing radiation dose is observed for the magnitude of micro-distortions while the change in the average crystallite sizes for the studied structures is minimal. The obtained dependences indicate that the greatest contribution to the change in structural parameters is made by deformation effects, as well as a change in their concentration as a result of irradiation.

One of the ways to reduce deformations and distortions in microstructures is the use of various types of ionizing radiation, the principle of which is to transfer the energy of the incident particles to the crystal structure of micromaterials due to elastic and inelastic interactions, which subsequently transforms into thermal and leads to a change in the size of crystallites, dislocation density, as well as partial annealing of defects due to an increase in the contribution of thermal vibrations and partial rearrangement of crystalline structures. It is worth noting that, in contrast to bulk samples, in which the transferred energy of incident particles is transformed into heat in a sufficiently large volume, as a result of which the processes of relaxation of defects are less pronounced. In the microstructures, all structural changes are limited by small sizes of objects not exceeding 350 to 400 nm. Previously, various research groups, including our group, have shown the promise of using various types of ionizing radiation to modify the properties of micromaterials, as well as assessing changes in structural and conductive properties [36,37,38,39,40,41,42]. In contrast to thermal annealing [43,44,45], which is based on a roughly similar principle of defect annealing due to a change in the magnitude of thermal vibrations, ionizing radiation allows modification pointwise inside microstructures [46,47]. In this case, the use of electronic radiation allows the modification of microstructures without a destructive effect.

According to the data obtained, for irradiated samples, the change in X-ray diffraction patterns can be divided into three main stages corresponding to different phase transformations. The first stage is typical for small doses of 50 to 150 kGy. At this stage, the change in X-ray diffraction patterns is characterized by a change in the shape of the diffraction lines, a decrease in the asymmetry of the peaks, and also a slight change in the intensities of the diffraction maxima. A change in the shapes and width of the diffraction peaks indicates a partial relaxation of distortions and strains in the crystal structure, which may be due to electronic annealing of defects. In this case, small doses of radiation lead to an insignificant change in the phase composition, as evidenced by the diagram shown in Figure 4a. The second stage of the change corresponds to the region with radiation doses of 200 to 250 kGy. At these irradiation doses, a sharp change in the phase composition is observed due to a decrease in the contribution of the FeNi_3_ phase, which indicates a rearrangement of the crystal structure as a result of irradiation. The dominance of the FeNi phase at high radiation doses is due to annealing of defects and distortions, as well as to processes of recrystallization as a result of reorientation and coarsening of crystallites, as well as a decrease in the dislocation and vacancy density of defects. The third stage is typical for radiation doses of 300 to 500 kGy. At these radiation doses, a sharp change in the diffraction pattern is observed, consisting in the absence of peaks characteristic of the FeNi_3_ phase and an increase in the intensity and changes in the shape of diffraction maxima for the FeNi phase. It should be noted that an increase in the radiation dose above 400 kGy does not lead to a significant change in the diffraction pattern, which may be due to saturation of the annealing process of defects in the structure.

Figure 4b presents the dynamics of changes in the degree of perfection of the crystal structure during irradiation, as well as changes in the density of microstructures. The degree of perfection of the crystal structure was estimated by analyzing diffraction lines using the pseudo-Voigt functions used to approximate the profile of X-ray peaks and assessing the concentration of disordered regions in the structure. According to the data obtained, the degree of perfection of the initial sample is 84%, which indicates a high degree of crystallinity of the obtained microstructures. An increase in the degree of crystallinity and density of microstructures as a result of irradiation indicates the electronic annealing of defects, as well as a change in the phase composition with an increase in the irradiation fluence. Figure 5 shows the dynamics of changes in the crystal lattice parameter, as well as the degree of distortion of the crystal lattice depending on the radiation dose. From the presented data, it can be seen that the presence of the second phase of FeNi_3_ leads to severe deformation of the crystal lattice and its distortion, as well as a deviation of the parameters from the reference value. Moreover, a decrease in the contribution of the FeNi_3_ phase leads to an ordering of the crystal structure, as well as a decrease in distorting factors. It should be noted that an increase in the irradiation dose above 350 kGy does not lead to a significant change in the crystal lattice parameters, which also confirms the assumption expressed above about the saturation of the annealing effect. Based on the data obtained, it can be concluded that the FeNi_3_ phase makes a large contribution to the deformation of the structure, a decrease in the contribution of which leads to the ordering of the crystal structure.

The most common types of defects in microstructural materials are vacancy and dislocation defects in the crystal structure, the presence of which is due to the processes of obtaining microstructures. As a rule, the synthesis of microstructures by electrochemical deposition is accompanied by nonequilibrium nucleation processes with a large release of hydrogen or other impurity inclusions capable of being introduced into the nodes and internodes of the crystal structure with its subsequent distortion. Moreover, the high density of dislocations and vacancies in the crystal structure can have a negative effect not only on strength properties but also on conductive ones. One of the ways to reduce the dislocation and vacancy density of defects, as mentioned above, is to increase thermal vibrations due to the transfer of thermal energy to the crystal structure with the subsequent annihilation of a large number of defects. In the case of irradiation with electrons with an energy of 5 MeV, the energy transferred as a result of elastic and inelastic collisions is 200 to 250 keV. This amount of energy is enough for the formation of highly mobile point defects that can migrate along the structure with the subsequent annihilation of structural defects. Figure 6 shows the dynamics of changes in the concentration of vacancy and dislocation densities as a result of irradiation.

According to the presented dependences of changes in the dislocation and vacancy densities, with an increase in the radiation dose, a decrease in these values is observed, which indicates electronic annealing of defects and improvement of the crystal structure.

The most important characteristic of the applicability of microstructures in conditions where they are exposed to external influences is their resistance to oxidation and degradation in media, as well as the oxidation rate of microstructures. The acidity of the medium is determined by the concentration of H^+^ and OH^−^ ions, which are introduced into the structure of microtubes, with subsequent destruction chemical and crystalline bonds due to the formation of impurity inclusions. To study the resistance of the synthesized microstructures to the effects of an aggressive environment, as well as assess the applicability of the electronic modification to increase the corrosion resistance, five types of microtubes were selected: Initial and irradiated with a dose of 100, 250, 400, and 500 kGy. The choice of doses was determined by the phase composition of the obtained microstructures after modification. A strongly acidic aqueous solution with pH = 1 was used as a model medium. Assessment of corrosion resistance was carried out by constructing anamorphoses of kinetic reaction curves, as well as by determining the rate of degradation of the microtubes in solution. Figure 7a shows the dynamics of changes in the obtained anamorphoses. The rate of the oxidation reaction of the studied structures was determined by assessing the change in the concentration of phases, as well as structural parameters, such as the degree of perfection and density per unit time. To determine the order of the chemical reaction of the oxidation of microtubes depending on the time spent in the medium, an integral method for determining the kinetic curves was used. Figure 7c shows X-ray diffraction patterns after 20 days of testing in an aggressive environment. 

According to the obtained kinetic curves, it was found that the initial and irradiated microtubes with a dose of 100 kGy are subject to the greatest degradation of structural properties. The high degradation rate for these samples is due to strong distortions of the crystal structure, the presence of a high density of dislocation and vacancy defects, as well as the phase composition. The presence of two phases leads to the appearance in the structure of differently oriented crystallites of different sizes, which contributes to a large number of grain boundaries, as well as a decrease in density. As a result, when the medium interacts with the surface of the microtubes, the degradation processes occur more intensively due to the migration of oxygen and hydrogen along the grain boundaries into the depths of the crystal lattice, with subsequent introduction into the nodes and interstices. The introduced oxygen and hydrogen can lead to the formation of stable oxide and hydroxide complexes and impurity inclusions with iron or nickel. It is worth noting that oxidation processes occur mainly during the formation of iron oxides and hydroxides, such as FeO/Fe(OH)_2_, while nickel oxidation is accompanied by the formation of unstable compounds, Ni_2_O_3_ and NiOOH, which fall apart into components. In this case, iron and hydroxide inclusions of iron are stable and are identified by X-ray diffraction after 10 days in the medium for the initial microstructures. For modified samples, in the structure of which the concentration of dislocation and vacancy defects is reduced by a factor of 3, the rate of degradation is minimal. This is due to the fact that in the case of irradiation, most of the defects in the structure of the microtubes are annihilated, resulting in an increase in crystallinity and a decrease in the integral porosity of the crystal structure. In turn, the absence of a second phase and an increase in crystallites for the modified samples leads to a decrease in the number of grain boundaries along which oxygen and hydrogen can be introduced from the medium. It is also worth noting that the low rate of degradation leads to the appearance of stable oxide compounds with iron only for 18 to 20 days (see the dependence data in Figure 7b and the X-ray diffraction pattern in Figure 7c). Figure 8 shows SEM images of the studied microtubes in an aggressive environment for 20 days.

For the initial microtubule samples irradiated and irradiated with a dose of 100 kGy, the corrosion processes are accompanied by the formation of ulcerative porous inclusions along the entire length of the microtubes while sections with a high content of ulcer inclusions are found, which indicates that corrosion processes occur near a large accumulation of grain boundaries. Also, on the surface of degraded samples, the formation of feather growth is observed, the formation of which is associated with oxidation processes. In the case of a small content of the FeNi_3_ phase in the structure of microtubes for samples irradiated with a dose of 250 kGy, only growths are observed on the surface of microtubes, without visible ulcerative inclusions. Based on the data obtained, it can be concluded that the appearance of ulcerative inclusions can be due to the presence of the FeNi_3_ phase, which is unstable and contains a large amount of nickel, the oxide compounds of which are unstable and can lead to partial destruction of crystalline and chemical bonds. Unlike the original samples, for modified samples with a dose of 400 to 500 kGy, the appearance of ulcerative inclusions is not observed, which indicates a high resistance to oxidation and degradation.

Thus, the study shows the promise of the use of electron irradiation with doses above 250 kGy for the directed modification of FeNi nanotubes by changing the phase composition, as well as reducing defective inclusions in the structure.

## 4. Conclusions

The work was devoted to the study of phase transition processes and structural changes in FeNi nanostructures exposed to electron radiation with an energy of 5 MeV and doses of 50 to 500 kGy. In the course of the study, the dose dependence of the change in the phase composition and structural characteristics of the studied samples was established. SEM, EDA, and SAR were used as research methods. Using the method of X-ray diffraction, three stages of phase transformations were established: FeNi_3_/FeNi→FeNi_3_ ≪ FeNi→FeNi. It was found that an increase in the radiation dose above 400 kGy does not lead to a significant change in the diffraction pattern, which is due to the saturation of the annealing process of defects in the structure. According to the dependences of changes in the dislocation and vacancy densities, it was found that with an increase in the radiation dose, a decrease in these values was observed, which indicates electronic annealing of defects and an improvement in the crystal structure, as well as an increase in the degree of crystallinity. During corrosion tests, it was found that for unmodified nanostructures, the main mechanism of corrosion is associated with the formation of ulcerative inclusions and the appearance of oxide and hydroxide inclusions. Unlike the original samples, for modified samples with a dose of 400 to 500 kGy, the appearance of ulcerative inclusions was not observed, which indicates a high resistance to oxidation and degradation.

## Figures and Tables

**Figure 1 nanomaterials-10-00047-f001:**
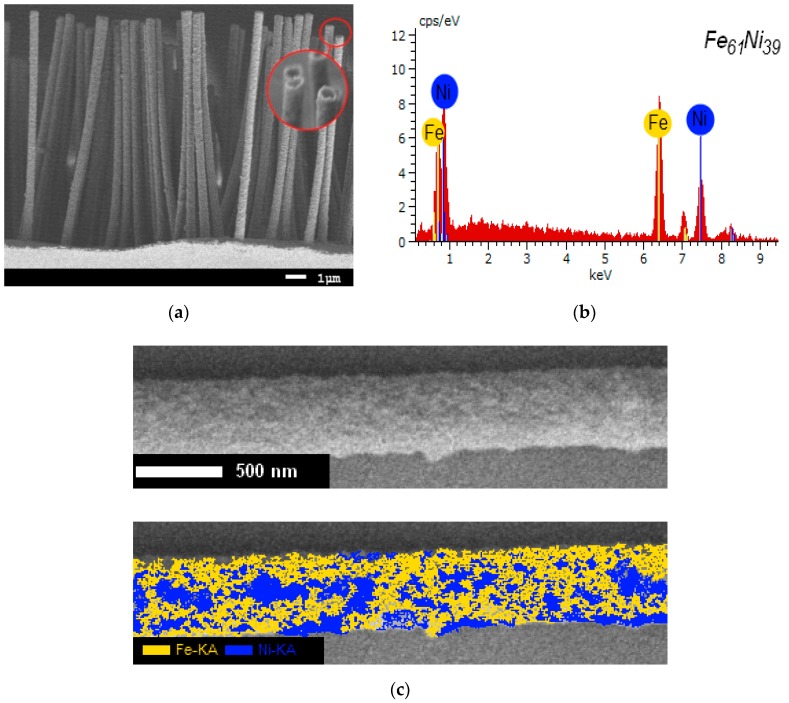
(**a**) SEM image of an array of microstructures; (**b**) energy dispersive analysis data; (**c**) mapping results; (**d**) the Mössbauer spectrum and the result of reconstructing the distribution of the hyperfine magnetic field of the studied microstructures.

**Figure 2 nanomaterials-10-00047-f002:**
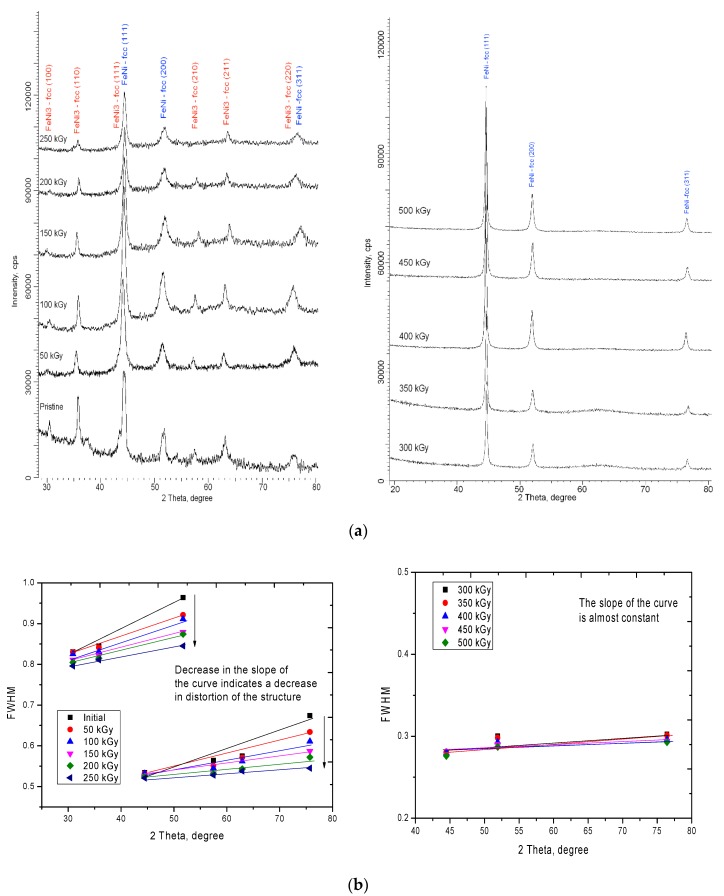
(**a**) X-ray diffraction patterns of the studied microstructures before and after irradiation; (**b**) Construction of Williamson–Hall for the studied microstructures depending on the dose of radiation.

**Figure 3 nanomaterials-10-00047-f003:**
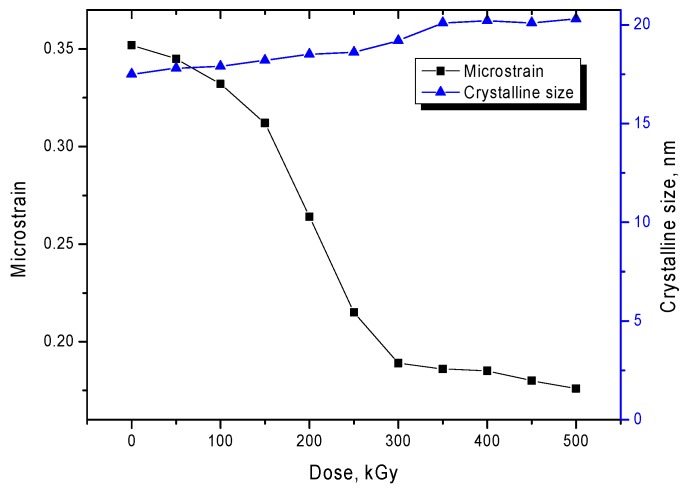
A graph of the dependence of changes in micro distortions and crystallite sizes in the structure of microtubes.

**Figure 4 nanomaterials-10-00047-f004:**
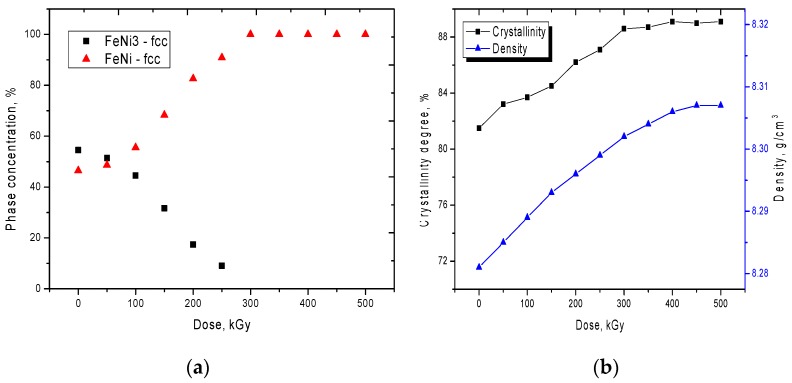
(**a**) Diagram of phase transformations in microstructures as a result of irradiation; (**b**) A plot of changes in crystallinity and density versus radiation dose.

**Figure 5 nanomaterials-10-00047-f005:**
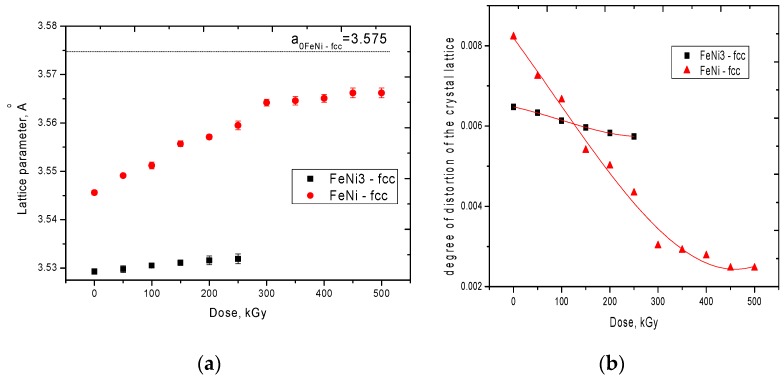
(**a**) Dynamics of changes in the crystal lattice parameters as a result of irradiation; (**b**) Graph of the dependence of the degree of disorder of the crystal lattice on the radiation dose.

**Figure 6 nanomaterials-10-00047-f006:**
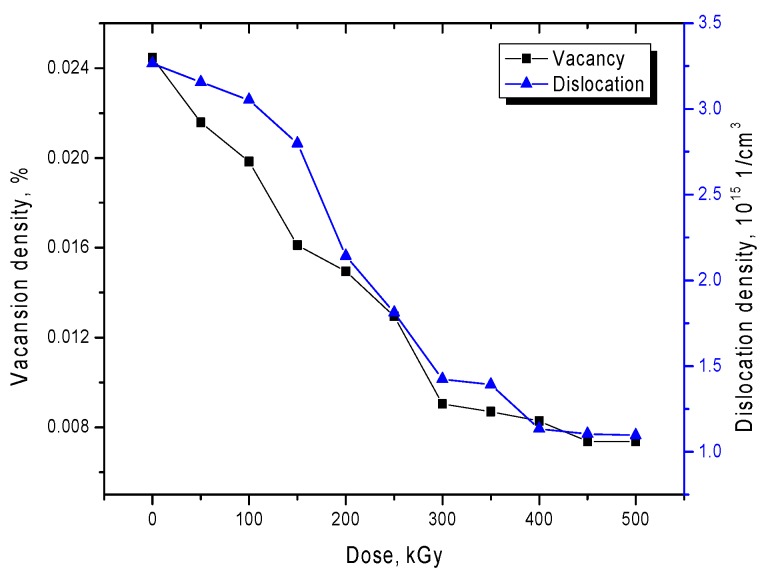
Dynamics of changes in vacancy and dislocation densities in the structure of microtubes.

**Figure 7 nanomaterials-10-00047-f007:**
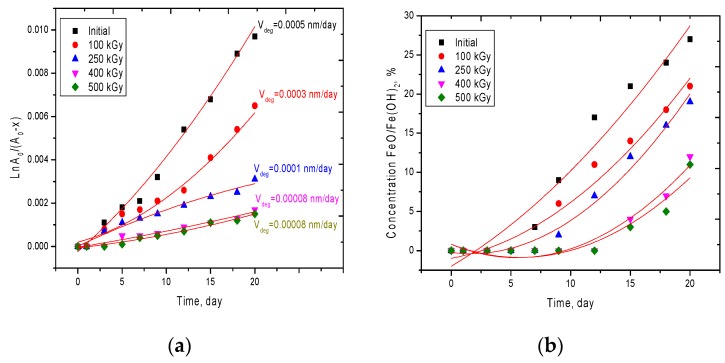
(**a**) Graph of the dynamics of changes in the anamorphoses of the kinetic curve for the degradation reactions of the studied microstructures; (**b**) A graph of the concentration of oxide phases in the structure of microtubes on the time of degradation; (**c**) X-ray diffraction patterns of the test samples after 20 days in an aggressive environment.

**Figure 8 nanomaterials-10-00047-f008:**
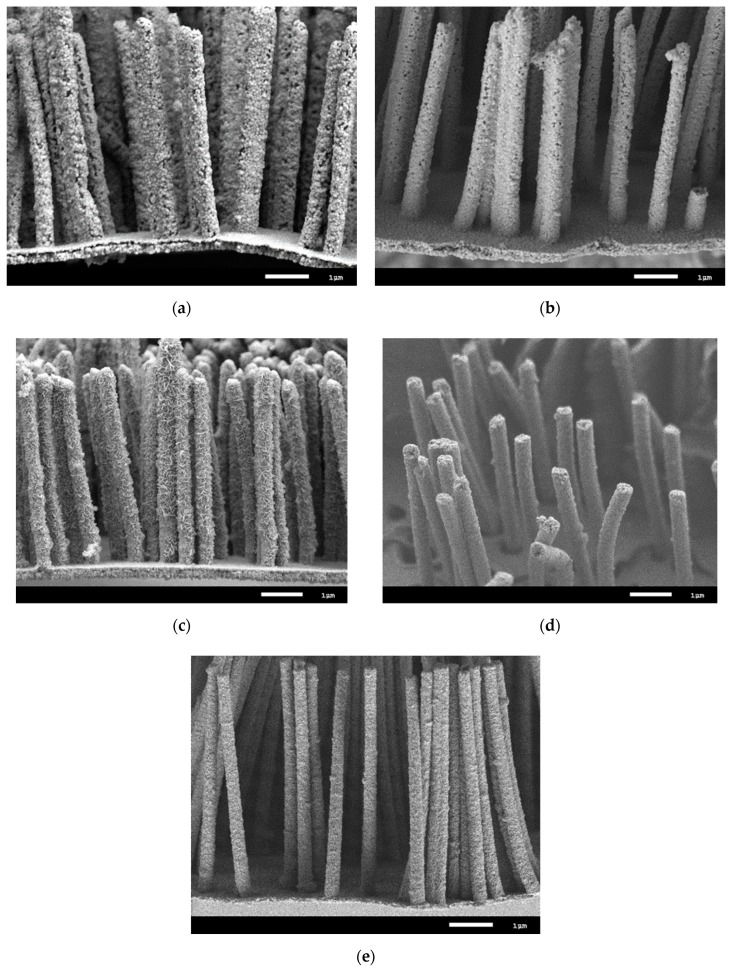
SEM images of the studied microtubes after degradation.: (**a**) Initial (**b**) 100 kGy (**c**) 250 kGy; (**d**) 400 kGy; (**e**) 500 kGy.

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
