# Peer review of "The Study of the Applicability of Electron Irradiation for FeNi Microtubes Modification"

_nanomaterials, 2019, doi:10.3390/nano10010047_

Round 1

Reviewer 1 Report

The background is opened well enough for a reader. For instance, the reasons to use nanomaterials with multiple boundaries as possible sinks (drains) for radiation defects (line 55) are smartly proposed.

The language of the paper is quite correct, but, at the same time, difficult in terms of syntax and is thus somewhat complicated to read. The sentences may be just too long and not perfectly structured. For instance, it is said in the abstract (line 22) that „It was found that an increase in the radiation dose above 400  kGy does not lead to a significant change in the diffraction pattern, which is due to the saturation of the annealing process of defects in the structure.“ It took some time to understand what the authors actually did want to say and why they use the expression „does not lead“. They probably meant something approximately like that: „After rising the radiation dose above 400 kGy, no further phase changes were followed, indicating the saturation of defect annealing and completion of the lattice formation process“

The figures are mainly quite clear, except the quite important Figure 2 with XRD patterns. The axes and patterns themselves do not look visually very distinct, neither do the reproduce with utmost clarity in outprint. The letters on both axes and indexes should also be larger. Application of colours in such figures is not relevant/necessary, the main target is the clarity of the peaks and lines.

Finally, there is a terminological issue. The objects the authors deal with are not nanoobjects. The FeNi nanotubes here are actually microtubes. A nanodimensional sample pre-requires its one characteristic dimension be below 100 nm, according to IUPAC criteria. With these nanotubes in this paper, even the diameter of the tubes look larger than 200-400 nm. In practice, the boundary between nano- and micro-objects is rather hazy, this can also be understood and taken into account.

Actually, the abovementioned issues do not necessarily reduce the scientific quality of the paper. The content of the paper is certainly interesting and deserves publication. On the other hand, the quality of the presentation could be improved.

Author Response

The authors are grateful to the referee for their comments on this work.

The text of the article has been amended according to the comments of the reviewer, as well as additional experimental data and a description of the results. All changes are highlighted in yellow in the text of the article.

Reviewer 2 Report

General comment

The work of Borgekov et al. investigates the effect of electron irradiation on the modification of FeNi nanotubes by changing the structural composition. This topic and the related contents are of sufficient interest for Nanomaterials, but, to my opinion, the analysis and interpretation of the results remain too qualitative or not always supported by the data. The authors should use also electron diffraction of X-ray scattering experiments on single-crystal to confirm their assumptions on crystallinity, defects and vacancy of their samples. Therefore, I recommend that the manuscript should be re-examined by the authors before publication in Nanomaterials.

Additional comments

The end of introduction may be developed to allow the reader to get the methodology and main results of the study Since X-ray measurements are reported in 2theta values, the wavelength used for XRD characterization should be mentioned. Line 112: Comma instead of dot in reference [32,33] Line 126-132: Mössbauer experiments are not described in the Method Section. Furthermore, providing a figure with Mössbauer data will help the reader and may strengthen the manuscript. Line 142: The assumption that SEM observations confirm the high degree of crystallinity and good strength properties of the samples is not obvious. In fact, how the authors can determine mechanical properties of their nanotubes only on electronic observations? Figure 1c: is it a chemical mapping along one single nanotube? If so, the uniformity as argued in Line 155, should be revised since the chemical mapping presents high-rich region of Ni or Fe. A comment on this feature? In several places, dynamics of changes (in X-ray diffraction, degree of perfection of the crystals) is used. If it is the case, the dynamic constant (rate of the reaction…) should be provided. Line 163: The peaks are not asymmetric at all. It seems however the presence of two phases may “deform” the shape of peak. The authors should clarify this point by providing quantitative analysis of the diffractograms. Futhermore, it is stated that Rietveld refinement has been performed. The results of such refinement should be presented in Figure 2. Line 179: how the authors calculate the average size (to be mentioned somewhere)? Is it corresponds to the average correlation length of crystallite size inside nanotubes? Line 228: The use of Pseudo-Voigt function is a qualitative method for line-shape analysis, but without any physical meaning for quantitative X-ray analysis. I suggest that in-depth analysis of X-ray data from Rietveld refinement. In Figure 6, what are the parameters involved in the kinetics curves? The authors should provide the model used to fit the data. Additional characterization of acid-treated samples should be provided to complete SEM observations (XRD for instance?).

Author Response

The authors are grateful for all comments and appreciation of their work. Please see the attachment.

Round 2

Reviewer 1 Report

The referee's comments have been addressed and the paper can be published.

Reviewer 2 Report

The revised manuscript accounts for my remaining suggestions from the previous round of review. I recommend publication in Nanomaterials.